# EGFRvIII Promotes the Proneural–Mesenchymal Transition of Glioblastoma Multiforme and Reduces Its Sensitivity to Temozolomide by Regulating the NF-κB/ALDH1A3 Axis

**DOI:** 10.3390/genes14030651

**Published:** 2023-03-04

**Authors:** Zhong-Fang Shi, Guan-Zhang Li, You Zhai, Chang-Qing Pan, Di Wang, Ming-Chen Yu, Chi Liu, Wei Zhang, Xiao-Guang Yu

**Affiliations:** 1Department of Biochemistry & Molecular Biology, Harbin Medical University, Harbin 150081, China; 2Department of Pathophysiology, Beijing Neurosurgical Institute, Beijing Tiantan Hospital, Capital Medical University, Beijing 100070, China; 3Department of Neurosurgery, Beijing Tiantan Hospital, Capital Medical University, Beijing 100070, China

**Keywords:** glioblastoma, EGFRvIII, ALDH1A3, NF-κB, proneural–mesenchymal transition, temozolomide

## Abstract

(1) Background: Glioblastoma multiforme (GBM) is the most common and malignant intracranial tumor in adults. At present, temozolomide (TMZ) is recognized as the preferred chemotherapeutic drug for GBM, but some patients have low sensitivity to TMZ or chemotherapy resistance to TMZ. Our previous study found that GBM patients with EGFRvIII (+) have low sensitivity to TMZ. However, the reasons and possible mechanisms of the chemoradiotherapy resistance in GBM patients with EGFRvIII (+) are not clear. (2) Methods: In this study, tissue samples of patients with GBM, GBM cell lines, glioma stem cell lines, and NSG mice were used to explore the causes and possible mechanisms of low sensitivity to TMZ in patients with EGFRvIII (+)-GBM. (3) Results: The study found that EGFRvIII promoted the proneural–mesenchymal transition of GBM and reduced its sensitivity to TMZ, and EGFRvIII regulated of the expression of ALDH1A3. (4) Conclusions: EGFRvIII activated the NF-κB pathway and further regulated the expression of ALDH1A3 to promote the proneural–mesenchymal transition of GBM and reduce its sensitivity to TMZ, which will provide an experimental basis for the selection of clinical drugs for GBM patients with EGFRvIII (+).

## 1. Introduction

Glioma is the most common intracranial malignant tumor in adults (accounting for 40–60%), and more than half of them are glioblastoma multiform (GBM), with the highest degree of malignancy [1]. GBM is mainly characterized by diffuse and invasive growth, which is difficult to completely remove by surgery and prone to recurrence after surgery [2], as well as having a poor prognosis. The median survival period is only 14 months [3]. At present, the standard first-line chemotherapy drug for GBM is temozolomide (TMZ). Some patients have low sensitivity to TMZ chemotherapy or chemotherapy resistance [4,5,6]. At present, a number of studies have suggested that the mechanism of TMZ resistance is very complex [7,8]. GBM can be divided into proneural (PN), neuronal (neural), classical (classical), and mesenchymal (MES) subtypes [9]. Recently, on the basis of transcriptional signatures, GBMs can be classified into three subtypes: PN, classical, and MES subtypes [10]. When the PN subtype GBM relapses, it undergoes the proneural–mesenchymal transition (PMT) [11,12,13] as to obtain a stronger proliferation ability and radiochemotherapy resistance. MES GBM has the characteristics of glioma stem cells. The appearance of glioma stem cells is considered to be one of the reasons for recurrence and drug resistance of GBM [1]. Studies have shown that glioma stem cells are a kind of cell with self-renewal, differentiation potential, and infinite proliferation, which are considered to be one of the important reasons for the occurrence and development of GBM and the resistance to chemoradiotherapy [14,15].

Epidermal growth factor receptor (EGFR) is a transmembrane protein that is highly expressed in lung cancer, liver cancer, bladder cancer, breast cancer, and other malignant tumors [16]. The expression of EGFR appears in 25–40% of GBM patients [17]. It has been reported that patients with GBM expressing EGFR have poor prognosis [18] and often have *EGFR* gene dislocation, deletion and mutation. The most common mutant of EGFR is EGFRvIII. It has been reported that EGFRvIII is not expressed in normal brain tissue [19], while about 50% of patients with GBM expressing EGFR have EGFRvIII (+) [6], and EGFRvIII (+) is closely related to the poor prognosis of GBM patients and chemoradiotherapy resistance [20,21,22]. In this study, it was found that patients with GBM EGFRvIII (−) were sensitive to TMZ chemotherapy, while patients with GBM EGFRvIII (+) had low sensitivity to TMZ, that is to say, patients with GBM EGFRvIII (+) had TMZ chemotherapy resistance. However, the reason for the low sensitivity to TMZ in the patients with GBM EGFRvIII (+) is not clear.

Aldehyde dehydrogenases (ALDHs) are a group of enzymes that metabolize endogenous (lipid, amino acids, or vitamins) and exogenous (drugs and ethanol) aldehydes to their corresponding carboxylic acids [23]. ALDH activity is crucial for cell differentiation, detoxification, and drug resistance. ALDHs contain 19 different subtypes, which are widely distributed in the body [24]. Among them, ALDH1A3, one of the ALDH1 subtypes, is located on chromosome 15q26.3 and is abnormally expressed in various tumor tissues, participating in regulating the biological function of tumor cells. It is reported that the methylation level of *ALDH1A3* promoter region is closely related to the survival of patients with primary GBM [25]. The high expression of ALDH1A3 is an important reason for the radiotherapy resistance of glioma stem cells [26]; ALDH1A3 is highly expressed in MES GBM and can be used as a marker of MES GBM [27]. Whether ALDH1A3 is regulated by EGFRvIII to reduce the sensitivity to TMZ in the GBM patients with EGFRvIII (+) has not been reported.

Nuclear factor-κB (NF-κB) exists in almost all cells and participates in the regulation of many important cell functions [28]. Numerous studies have confirmed that NF-κB is expressed in a variety of tumors, and its expression level is related to the degree of malignancy of the tumor [29]. Research shows that the NF-κB pathway is over-activated in GBM and is involved in cell proliferation, invasion, and radiochemotherapy resistance of GBM [30,31]. Some researchers found that mTORC2 passes NF-κB-mediated cisplatin resistance in GBM [32]; the EGFRvIII/PI3K/AKT/NF-κB pathway participates in TMZ resistance by regulating the key gene *E2F6* of GBM drug resistance [16]. Other studies have reported that NF-κB inhibitor can downregulate the activity of *ALDH1A3* promoter and then inhibit the transcription of *ALDH1A3* in pleural mesothelioma cell lines [33]. It has not been reported whether NF-κB regulates the transcription of *ALDH1A3* in GBM cells and whether EGFRvIII regulates the expression of ALDH1A3 through NF-κB.

This study intended to explore the relationship between EGFRvIII and GBM in patients with GBM, GBM cell lines, glioma stem cell lines, and NSG mice; the effect of EGFRvIII on the proneural–mesenchymal transition of GBM and glioma stem cells and the chemosensitivity to TMZ; and further screening and identification of target genes and signal pathways downstream of EGFRvIII. The mechanism of EGFRvIII affecting the proneural–mesenchymal transition and chemosensitivity of TMZ are also discussed. This study provides an experimental basis for the clinical drug selection of patients with GBM EGFRvIII (+).

## 2. Materials and Methods

### 2.1. Patients and Samples

The collection of samples and clinical information in this study was approved by the Institutional Review Board (IRB) of the Beijing Tiantan Hospital (KY2014-002-02). Written consent was obtained from each patient. At least two neuropathologists diagnosed each sample in this study as GBM. The patients were followed up every 3 months by phone calls or clinical visits. All patients’ clinical and follow-up information in this study is available on the Chinese Glioma Genome Atlas (CGGA) website (http://www.cgga.org.cn/) accessed on 1 January 2019. Transcriptome microarray and clinical data of glioma samples were from CGGA generating with Agilent platform (Illumina Solexa). Overall survival was estimated from the date of diagnosis to the date of either death or last follow-up.

### 2.2. Cell Culture

GBM cell lines LN229 and U87 were obtained from the Institute of Biochemistry and Cell Biology, Chinese Academy of Science. All GBM cell lines were identified by STR Profiling and cultured in Dulbecco’s modified Eagle’s medium (DMEM) (Gibco, NY, USA) supplemented with 10% fetal bovine serum (FBS) (Gibco, NY, USA) and 1% penicillin–streptomycin (Gibco, NY, USA).

The human glioma stem cell line (BNI-21-1-S) was used in this study and was acquired from patient specimens at Beijing Tiantan Hospital under protocols approved by the Institutional Review Board (IRB) and maintained as tumorspheres in GSC media. BNI-21-1-S cells were maintained in a 5% CO_2_ atmosphere at 37 °C in the serum-free medium containing DMEM/F12 (Gibco, NY, USA) supplemented with B27 (Gibco, NY, USA), basic fibroblast growth factor (bFGF, 20 ng/mL), epidermal growth factor (EGF, 20 ng/mL), and heparin (2.5 mg/mL). Growth factors (bFGF and EGF) were added twice a week. BNI-21-1-S cells were enzymatically dissociated into single cells using Accutase (Sigma-Aldrich, St. Louis, MO, USA) and thereafter routinely cultured in the serum free medium every 4–6 days.

The EGFRvIII stable GBM cell lines and glioma stem cell line were obtained using lentiviral transfection technology as previously described [34] and were treated with TMZ at the indicated concentrations for 72 h, and then cellular viabilities were determined by the CCK8 assay.

### 2.3. Cell Proliferation and Transwell Migration Assays

Cell proliferation was measured using the Cell Counting Kit-8 (CCK-8, Dojindo Laboratories, Kumamoto, Japan). Specifically, 2.0 × 10^3^ cells/well were seeded into six pairs of duplicate wells of a 96-well plate and incubated at 37 °C. At the indicated time point, the medium for six-repeated wells was changed, and 10 μL CCK-8 reagent was added to the 100 μL fresh culture medium. Absorbance was detected daily at 450 nm.

For migration assays, GBM cells (1 × 10^5^) in serum-free DMEM were added to the Transwell upper chamber, with the lower chamber containing DMEM with 10% FBS. After incubation for 6 h at 37 °C, the migrated cells present on the underside of the Transwell membrane were stained with crystal violet and counted under a microscope (Carl Zeiss Meditec AG, Jena, Germany).

### 2.4. Real-Time Quantitative Reverse Transcription Polymerase Chain Reaction (RT-qPCR)

Total RNA was extracted using a Total RNA Extraction Kit (Promega, Madison, WI, USA) according to the manufacturer’s instructions. RNA intensity was assessed using a NanoDrop 2000 (Agilent Technologies, Santa Clara, CA, USA). cDNA was generated from 1 μg of total RNA using a Reverse Transcription System (Promega Corporation, Wisconsin, DC, USA). The ABI 7500 Real-time PCR System analyzed expression levels of target genes. The samples were treated at 95 °C for 10 min followed by 40 cycles at 95 °C for 15 s, 58 °C for 30 s, and 72 °C for 30 s, and a final extension at 72 °C for 10 min. GAPDH was used as the internal control. The relative mRNA expression levels of target genes were calculated by the 2^−ΔΔCt^ method. The primer sequences for various human genes used in this study were listed were listed as follows: *ALDH1A3*-forward: TGAATGGCACGAATCCAAGAG; *ALDH1A3*-reverse: CACGTCGGGCTTATCTCCT; *BCL2A1*-forward: ATGGATAAGGCAAAACGGAG; *BCL2A1*-reverse: TGGAGTGTCCTTTCTGGTCA; *CD44*-forward: CCCAGATGGAGAAAGCTCTG; *CD44*-reverse: ACTTGGCTTTCTGTCCTCCA; *CD133*-forward: ACTCCCATAAAGCTGGACCC; *CD133*-reverse: ATTTTGGATTCATATGCCTT; *EGFRvIII*-forward: GGCTGGGTAGGGTAGAGATTCC; *EGFRvIII*-reverse: GCTGAACTGCTGGTGAGGATGA; *LYN*-forward: CTGAACTCAAGTCACCGTGG; *LYN*-reverse: TCATGCTTAGTCCACTGTCTGT; *NOTCH1*-forward: GAGGCGTGGCAGACTATGC; *NOTCH1*-reverse: CTTGTACTCCGTCAGCGTGA; *OLIG2*-forward: CTCCTCAAATCGCATCCAGA; *OLIG2*-reverse: AGAAAAAGGTCATCGGGCTCTCA; *WT1*-forward: TTAAAGGGAGTTGCTGCTGG; *WT1*-reverse: GACACCGTGCGTGTGTATTC; *SOX2*-forward: ACCGGCGGCAACCAGAAGAACAG; *SOX2*-reverse: GCGCCGCGGCCGGTATTTAT; *NF-kB*-forward: ACCAGGCTGGCAGCTCTTCT; *NF-kB*-reverse: GCGTGGTTCCAGGCACAACT; *GAPDH*-forward: GGAGCGAGATCCCTCCAAAAT; *GAPDH*-reverse: GGCTGTTGTCATACTTCTCATGG.

### 2.5. Western Blotting

Western blotting was performed as previously described [35]. After washing with ice-cold PBS three times, cells were lysed in ice-cold lysis buffer. A Bicinchoninic Acid Protein Assay kit (Thermo Fisher Scientific, Waltham, MA, USA) was used to measure total protein content. The proteins were boiled and then separated by 10% sodium dodecyl sulfate polyacrylamide gel electrophoresis (SDS-PAGE). After membrane transfer, PVDF membranes were incubated with primary antibodies targeting EGFRvIII (Cell Signaling Technology, Danvers, MA, USA, 1:1000), ALDH1A3 (ProteinTech Group, Chicago, IL, USA, 1:1000), CD44 (ProteinTech Group, Chicago, IL, USA, 1:1000), SOX2 (ProteinTech Group, Chicago, IL, USA, 1:1000), NF-κB p65 (Cell Signaling Technology, Danvers, MA, USA, 1:1000), phospho-NF-κB p65 (Cell Signaling Technology, Danvers, MA, USA, 1:1000), or β-actin (Sigma-Aldrich, St. Louis, MO, USA, 1:2000) in blocking solution overnight at 4 °C. Secondary goat anti-rabbit or goat anti-mouse antibodies conjugated with a horseradish peroxidase were incubated for 1 h at room temperature. An enhanced chemiluminescence reagent kit (Applygen Technologies, Beijing, China) was used to detect the specific bands, which were quantified with a ChemiDocMP (Bio-Rad, Hercules, CA, USA). The levels of targe protein expression were normalized to β-actin. The p-NF-κB levels were normalized to total NF-κB.

### 2.6. Chromatin Immunoprecipitation (ChIP)

A ChIP-IT High Sensitivity kit (Active Motif, Carlsbad, CA, USA) was used to determine the association of transcription factor p65 NF-κB with its specific genomic regions on the *ALDH1A3* gene. U87-EGFRvIII (+) cells were treated with 40 nmol/L of YM155 or 0.1% DMSO (control) for 24 h and subjected to cell fixation, chromatin sonication, immunoprecipitation, and DNA purification. Rabbit IgG was used as a negative control in the immunoprecipitation experiments. The immunoprecipitated fraction was analyzed by PCR and qRT-PCR to determine the abundance of the target DNA sequence(s) relative to the input chromatin. *ALDH1A3*-forward: CCTCCGGGGCGAGAGAGC; *ALDH1A3*-reverse: CGCCTAATTGGCTGCGGG.

### 2.7. Immunohistochemistry and Immunofluorescence

Immunohistochemistry was performed as previously reported [35]. Briefly, sections were deparaffinized and boiled in Tris antigen-retrieval buffer. Then, sections were incubated with primary antibodies against EGFRvIII (Cell Signaling Technology, Danvers, MA, USA, 1:200), ALDH1A3 (ProteinTech Group, Chicago, IL, USA, 1:400), CD44 (ProteinTech Group, Chicago, IL, USA, 1:400), and SOX2 (ProteinTech Group, Chicago, IL, USA, 1:400) overnight at 4 °C. Immunohistochemistry images were captured using an Axio Imager 2 (Carl Zeiss Meditec AG, Jena, Germany).

Immunofluorescence staining was performed according to our previous description [28], and cultured cells were fixed in acetone for 10–15 min. Cells were blocked with 10% goat serum for 20 min and then incubated with primary antibodies against EGFRvIII (Cell Signaling Technology, Danvers, MA, USA, 1:200 dilution), ALDH1A3 (ProteinTech Group, Chicago, IL, USA, 1:200), CD44 (ProteinTech Group, Chicago, IL, USA, 1:200), and SOX2 (ProteinTech Group, Chicago, IL, USA, 1:200) overnight at 4 °C. Subsequently, after washing, the cultures were incubated with Alexa Fluor 488- or 546-conjugated secondary antibodies (Life Technologies, Carlsbad, CA, USA, 1:200), applied at room temperature (RT) for 1 h. Images were acquired under an inverted fluorescence microscope. The negative control was treated with PBS instead of the primary antibody.

### 2.8. In Vivo Xenograft Growth

All animal studies were carried out according to the guidelines of the Declaration of Helsinki and approved by the Animal Care and Use Committee at Beijing Neurosurgical Institute (No. 202101028). Female NOD Scid γ mice (NSG mouse, weight 18–22 g, 5 weeks old) (Charles Rivers, Beijing, China) were selected for the experiments. Dissociated U87 cells (5 × 10^5^ cells in 5 μL of PBS) transfected with an empty vector or EGFRvIII overexpression were stereotactically injected into the right frontal node of NSG mice. The inoculation position was 2 mm lateral and 2 mm posterior to the anterior fontanel. Tumor size was estimated from luciferase volume measurements. The mice were sacrificed when they exhibited disturbed activity or convulsion. The brains were obtained by surgical operation and embedded in paraffin.

### 2.9. Bioluminescence Imaging

To investigate the effect of EGFRvIII on the malignant phenotype of GBM in vivo, we injected luciferase-labeled U87 cells expressing an empty vector, an EGFRvIII overexpression vector, into the brains of NSG mice. Tumor progression was monitored by bioluminescence imaging on days 13, 24, and 27.

### 2.10. Transcriptome Sequencing

Transcriptome sequencing data and most molecular pathological information (MGMT status, Chromosome 1p/19q status, IDH status, and transcription subtypes) of glioma patients were obtained from the CGGA official website (www.cgga.org.cn) accessed on 1 January 2019. The EGFR status was obtained from the immunohistochemical staining of the above clinical samples. Bioinformatics analysis was performed through R 4.2.1 and GraphPad Prism 9 software.

The total RNA in GBM cell lines LN229-EGFRvIII (+) and U87-EGFRvIII (+) was extracted using the TRIzol regent (Invitrogen, Carlsbad, CA, USA) following the manufacturer’s instructions. mRNA was enriched, the first and second strands of cDNA were synthesized, and then the cDNA library was constructed and inspected. Transcriptome sequencing data of glioma samples were generated using the Illumina Hiseq platform. Finally, the reference sequence alignment, counting, inter-sample filtering, and normalization were carried out, and the analysis of gene/transcript differences between sample groups were performed to obtain results. The genes related to EGFRvIII are shown in Appendix A.

The most relevant genes of EGFRvIII, or a characteristic gene list of the cell cluster, were uploaded to the Database for Annotation, Visualization, and Integrated Discovery (DAVID, v6.8). The official gene symbol was selected as an identifier, and Homo sapiens was selected as the species. Finally, the Kyoto Encyclopedia of Genes and Genomes (KEGG) pathway analysis enrichment results were obtained. The top six results in ascending order of *p*-value (*p* < 0.05) were displayed in this study.

Transcription factors were predicted using LASAGNA-Search 2.0 online tool (https://biogrid-lasagna.engr.uconn.edu/lasagna_search/) accessed on 1 January 2020. p50 and p65 were selected as TF Model Input, and the promoter region (−1000 to 0 relative to TSS) of *ALDH1A3* was selected as Promoter Sequence Input [36].

### 2.11. Statistical Analysis

All statistical computations were performed with the statistical software environment R (version 4.2.1), SPSS statistical package (version 19), GraphPad Prism (version 9). All data were expressed as mean ± SD. Student’s *t*-test was used to compare differences between two groups. The Kaplan–Meier method with a two-sided log-rank test was used to compare the overall survival of patients between different groups. The unpaired *t*-test and Fisher’s exact test were used to analyze the correlation between EGFRvIII expression and clinical data in 66 patients with GBM. All experiments were repeated three times. A *p*-value of <0.05 was considered statistically significant.

## 3. Results

### 3.1. EGFRvIII Affected the Gene Expression Profile of GBM Patients, and GBM Patients with EGFRvIII (+) Had Low Sensitivity to TMZ

EGFRvIII is closely related to tumor occurrence, development, and prognosis [19]. In order to explore the correlation between GBM and EGFRvIII, this study first used transcriptome sequencing data in the CGGA database to analyze the tumor tissue gene expression profile of the two groups of GBM patients with EGFRvIII (−) and (+). The results showed that there were significant differences in the gene expression profile of tumor tissue between the two groups of GBM patients with EGFRvIII (−) and (+) (*p* < 0.05) and no significant difference in the distribution of important molecular typing markers (including IDH1/2 status, 1p/19q codeletion status, *MGMT* promoter methylation state) were found between the two groups of GBM patients with EGFRvIII (−) and (+) (*p* > 0.05) (Figure 1A). The results suggest that the different expression of EGFRvIII may lead to the different biological characteristics of GBM cells, which was not related to important molecular typing markers of GBM.

Studies have shown that EGFRvIII is not expressed in normal brain tissue and is associated with the prognosis of patients with GBM [37,38,39]. In order to confirm whether EGFRvIII is closely related to GBM, immunohistochemical techniques were used in this study to detect clinical tissue samples from 66 cases of human GBM from Beijing Tiantan Hospital, and the results showed that EGFRvIII expression was negative in 15 cases and was positive in 51 cases, and the positive expression rate of EGFRvIII was 77% (51/66) (Figure 1B). The results showed that the positive expression rate of EGFRvIII was relatively high in GBM tumors.

We then analyzed the clinical data of 66 patients with GBM and found that there was no correlation between EGFRvIII expression and age, sex, IDH1 mutation, 1p/19q codeletion, *MGMT* promoter methylation, radiotherapy, or chemotherapy in GBM patients (*p* > 0.05) (Table 1).

It has been reported that EGFRvIII affects the chemotherapy sensitivity to TMZ in GBM patients [3,40,41], but the results are still controversial. To investigate the relationship between EGFRvIII and chemotherapy sensitivity to TMZ in GBM patients, we followed up 15 GBM patients with EGFRvIII (−) and found that compared with patients who did not receive TMZ, TMZ could significantly prolong overall survival (OS) (*p* < 0.05). We also followed up 51 patients with GBM with EGFRvIII (+) and found no difference in overall survival between patients receiving or not receiving TMZ (*p* > 0.05) (Figure 1C). The results showed that GBM patients with EGFRvIII (−) were sensitive to TMZ chemotherapy, while GBM patients with EGFRvIII (+) were less sensitive to TMZ.

### 3.2. GBM Cells and Glioma Stem Cells with EGFRvIII (+) Significantly Increased Cell Viability and Cell Migration

Firstly, we constructed three cell lines of EGFRvIII (+) through genetic engineering technology in the EGFRvIII (−) GBM cell line (U87, LN229) and glioma stem cell line (BNI-21-1-S): U87-EGFRvIII (+), LN229-EGFRvIII (+), and BNI-21-1-S-EGFRvIII (+). The results of qRT-PCR and Western blot showed that EGFRvIII expression was significantly upregulated in all three cell lines compared to the control group (*** *p* < 0.001) (Figure 2A,B). The results suggested that the GBM cell lines (U87, LN229) and glioma stem cell lines (BNI-21-1-S) of EGFRvIII (+) were successfully constructed.

In order to investigate the effect of EGFRvIII on the proliferation of GBM cells and glioma stem cells, the above three EGFRvIII (+) cell lines were used in this study, and the cell proliferation was detected by CCK8 assay. The results showed that, compared with the control group, the cell viability of all three cell lines was significantly increased (*p* < 0.001) (Figure 2C). The results indicated that EGFRvIII could promote the proliferation of GBM cells and glioma stem cells.

Secondly, we constructed a mouse model of orthotopic xenograft by use of U87-EGFRvIII (+) cells. After 13, 24, and 27 days, GBM tumor growth in the brain of mice was observed by bioluminescence imaging. The results showed that, compared with the control group, there was GBM formation in the brain of mice on day 13 of the experiment, and the tumor volume increased significantly on day 24 and day 27 (*p* < 0.05 and *p* < 0.01, respectively) (Figure 2D). The results indicated that EGFRvIII could promote tumor growth in GBM.

The above in vivo and in vitro experiments confirmed that EGFRvIII could promote the growth of GBM cells and glioma stem cells.

In order to investigate the effect of EGFRvIII on the migration of GBM cells and glioma stem cells, the above three EGFRvIII (+) cell lines were used to detect cell migration by Transwell assay. The results showed a significant increase in the number of EGFRvIII (+) cells migrating compared to the control group (*p* < 0.001) (Figure 2E). The results indicated that EGFRvIII could promote the migration of GBM cells and glioma stem cells.

### 3.3. GBM Cells and Glioma Stem Cells with EGFRvIII (+) Significantly Decreased the Expression of PN GBM Markers, Increased the Expression of MES GBM Markers, and Increased the IC_50_ Value of TMZ

The proneural–mesenchymal transition is the main driving factor for malignant progression of GBM [9,18] and one of the important reasons for TMZ resistance [42]. Among them, SOX2, NOTCH1, OLIG2, and CD133 are markers of proneural subtype (PN), while CD44, BCL2A1, LYN, and WT1 are markers of interstitial subtype (MES) [26].

In order to investigate the effect of EGFRvIII on the proneural–mesenchymal transition of GBM cells and glioma stem cells, the above three cell lines of EGFRvIII (+) were used in this study. Firstly, qRT-PCR was used to detect the expression of markers related to the proneural–mesenchymal transition. The results showed that compared with the control group, mRNA expressions of PN GBM markers including *SOX2*, *NOTCH1*, *OLIG2*, and *CD133* in EGFRvIII (+) cells decreased, and MES GBM markers including *CD44*, *BCL2A1*, *LYN*, and *WT1* were increased (*p* < 0.05) (Figure 3A–C).

Western blot was used to detect the expression of PN marker SOX2 and MES marker CD44. The results showed that, compared with the control group, the expression of PN marker SOX2 decreased in EGFRvIII (+) cells, while the expression of MES marker CD44 increased (Figure 3D). The expressions of PN marker SOX2 and MES marker CD44 in three EGFRvIII (+) cells were also detected by cell immunofluorescence staining, and the same results as Western blot were obtained (Figure 3E). The results indicated that EGFRvIII could promote the proneural–mesenchymal transition of GBM and glioma stem cells.

The mice were sacrificed after 27 days in a mouse model of orthotopic xenograft, and the brain tissues were taken from the mice, with the expressions of SOX2 and CD44 detected by immunohistochemical staining. The results showed that the expression of the PN marker SOX2 was decreased in EGFRvIII (+) cells compared with the control group, while the expression of the MES marker CD44 was increased (Figure 3F). The results indicated that EGFRvIII could promote the proneural–mesenchymal transition of GBM.

To investigate the effect of EGFRvIII on the sensitivity of GBM cells and glioma stem cells to TMZ, GBM cell lines U87-EGFRvIII (+) and LN229-EGFRvIII (+) were pretreated with different concentrations of TMZ (0, 100, 200, 300, 400, 600, 800, 1200, 1600 μmol/L), and glioma stem cell line BNI-21-1-S-EGFRvIII (+) was pretreated with TMZ (0, 25, 50, 75, 100, 150, 200, 300, 400 μmol/L). The IC_50_ values of the three cell lines were detected by CCK8. The results showed that the IC_50_ values of TMZ in EGFRvIII (+) GBM cells and glioma stem cells were significantly increased compared with the control group (U87: 600 μmol/L vs. 1200 μmol/L; LN229: 700 μmol/L vs. 1200 μmol/L; BNI-21-1-S: 150 μmol/L vs. 345 μmol/L) (*p* < 0.05) (Figure 3G). The results indicated that EGFRvIII could reduce the sensitivity of GBM cells and glioma stem cells to TMZ.

### 3.4. ALDH1A3 Was Screened and Identified as the Downstream Target Gene of EGFRvIII

The mRNA of GBM cell lines LN229-EGFRvIII (+) and U87-EGFRvIII (+) was extracted for second-generation transcriptional sequencing. Bioinformatics analysis was carried out, and the results showed that, compared with the control group, 325 genes were upregulated and 556 genes were downregulated in LN229-EGFRvIII (+) cells, and there were 1909 upregulated genes and 3246 downregulated genes in U87-EGFRvIII (+) cells (Figure 4A).

Further, we analyzed the differentially upregulated or downregulated genes of the above two kinds of cells, and the results showed that 201 genes were downregulated and 78 genes were upregulated (including *ALDH1A3*) (Figure 4B). Studies have shown that ALDH1A3 plays an important role in the mesenchymal differentiation and malignant progression of GBM cells [43,44,45,46]. Our previous studies showed that ALDH1A3 is one of the key metabolic enzymes involved in the occurrence and development of GBM [25,27,47]. It is suggested that *ALDH1A3* may be the target gene downstream of EGFRvIII.

In order to further confirm whether *ALDH1A3* is a downstream target gene regulated by EGFRvIII, we used the above three cell lines of EGFRvIII (+) to detect the expression of ALDH1A3 by qRT-PCR and Western blot. The results showed that mRNA and protein expression of ALDH1A3 were significantly increased compared with the control group (**** *p* < 0.0001) (Figure 4C,D).

Then, we also found, compared with the control group, the expressions of EGFRvIII and ALDH1A3 detected by immunohistochemical staining were significantly increased in the GBM tumor tissue of a mouse model of orthotopic xenograft (Figure 4E).

These results indicated that *ALDH1A3* was a downstream target gene of EGFRvIII.

### 3.5. Screening and Determining That the Downstream Signaling Pathway of EGFRvIII Was Related to the NF-κB Pathway, p- NF-κB in EGFRvIII (+) Cells Was Increased, and NF-κB Bound to the ALDH1A3 Promoter Region

A variety of signaling pathways may be involved in proneural–mesenchymal transition in GBM [12,48,49,50] and TMZ chemotherapy resistance [5,40,51]. KEGG analysis was performed using transcriptional sequencing data from brain tumor tissues of GBM patients in the CGGA database, and the results showed that pathways related to EGFRvIII included cytokine–cytokine receptor interaction, chemical factor signaling pathway, NF-κB pathway, etc. (Figure 5A). Studies have reported that NF-κB is involved in GBM cell proliferation, migration, and invasion, as well as chemoradiotherapy resistance [28,52]. EGFRvIII affects tumor growth, angiogenesis, and TMZ chemotherapy resistance through the NF-κB pathway [16,53]. Therefore, NF-κB may be one of the most closely related pathways of EGFRvIII in GBM.

KEGG analysis was also performed on the second-generation transcriptome sequencing data to screen EGFRvIII-related pathways in GBM cell lines. The results showed that compared with the control group, the EGFRvIII-related pathways in U87-EGFRvIII (+) cells included cytokine–cytokine receptor interaction, tumor signaling pathway, PI3K-Akt pathway, etc., and in LN229-EGFRvIII (+) cells included cytokine–cytokine receptor interaction, tumor signaling pathway, TNF signaling pathway, etc. (Figure 5B).

Further, we analyzed the different signaling pathways of the above two kinds of cells by gene set variation analysis (GSVA). The results showed that 4 pathway enrichment scores decreased and 28 increased in U87-EGFRvIII (+) and LN229-EGFRvIII (+) cells, including the NF-κB pathway (Figure 5C).

These results suggest that EGFRvIII is closely related to the NF-κB pathway in GBM.

In order to determine whether EGFRvIII can activate the NF-κB pathway, we detected p-NF-κB and NF-κB protein expressions by Western blot in U87-EGFRvIII (+). Compared with the control group, U87-EGFRvIII (+) cells increased the expression of phosphorylated NF-κB (Figure 5D). The results indicated that EGFRvIII could activate the NF-κB pathway.

The inhibition of the NF-κB pathway can downregulate the activity of the promoter of *ALDH1A3* gene, thus inhibiting the transcription of *ALDH1A3* gene in pleural mesothelioma cell lines [33]. To confirm the transcriptional regulation of NF-κB on the *ALDH1A3* gene in GBM cells, we first used LASAGNA software and motif analysis to predict the binding site of NF-κB and the *ALDH1A3* gene promoter region (Figure 5E,F). Furthermore, the ChIP assay was used to detect whether NF-κB was bound to the promoter region of the *ALDH1A3* gene. The results showed that compared with the control group, the enrichment of NF-κB in the promoter region of the *ALDH1A3* gene was increased in U87-EGFRvIII (+) cells (*p* < 0.05) (Figure 5G). The results showed that the binding of NF-κB to the promoter region of the *ALDH1A3* gene exists in GBM EGFRvIII (+) cells, indicating that NF-κB can transcriptionally regulate *ALDH1A3*.

U87-EGFRvIII (+) cells were pretreated with the NF-κB (JSH-23, 20 μmol/L). In vitro and in vivo studies have confirmed that JSH-23 effectively inhibits NF-κB transcription activity in a variety of tumor cells or tissues [54]. Western blot was used to detect the expression of EGFRvIII and ALDH1A3. The results showed that compared with the EGFRvIII (+) group, the NF-κB inhibitor JSH-23 did not affect the expression of EGFRvIII, but significantly decreased the expression of ALDH1A3 (*p* < 0.05) (Figure 5H). The expression of EGFRvIII and ALDH1A3 in EGFRvIII (−) cells was not affected by administration of JSH-23 (*p* > 0.05) (Figure 5H). The results indicated that the NF-κB inhibitor could reverse the increased expression of ALDH1A3 induced by EGFRvIII.

### 3.6. Inhibition of NF-κB Decreased ALDH1A3 Expression of EGFRvIII (+) Cells, Reversed the Enhanced Cell Migration Ability, Decreased the Expression of PN GBM Marker (SOX2), Increased the Expression of MES GBM Marker (CD44), and Increased the IC50 of TMZ Caused by EGFRvIII (+)

To determine whether EGFRvIII affects GBM cell migration via the NF-κB pathway, we used U87-EGFRvIII (+) cells for the Transwell assay. The results showed that compared with the control group, administration of the NF-κB inhibitor JSH-23 reversed the increased number of cell migrations induced by EGFRvIII (*p* < 0.05) (Figure 6A). The results indicated that EGFRvIII could activate the NF-κB pathway to promote GBM cell migration.

In order to determine whether EGFRvIII affects the proneural–mesenchymal transition of GBM cells through the NF-κB pathway, Western blot was used to detect the expression of proneural–mesenchymal-transition-related markers (PN marker SOX2 and MES marker CD44). The results showed that the administration of NF-κB inhibitor JSH-23 reversed EGFRvIII (+)-induced decrease in the expression of PN marker SOX2 and increased the expression of MES marker CD44 in GBM cells compared with the control group (Figure 6B). The results indicated that EGFRvIII could activate the NF-κB pathway to promote proneural–mesenchymal transition in GBM.

To determine whether EGFRvIII affects GBM cell sensitivity to TMZ via the NF-κB pathway, IC_50_ of U87-EGFRvIII (+) cells against TMZ was detected using CCK8 assay. The results showed that compared with the control group, the increase in IC_50_ value induced by EGFRvIII (+) could be reversed after the administration of the NF-κB inhibitor JSH-23 (*p* < 0.05) (Figure 6C). The results indicated that EGFRvIII could activate the NF-κB pathway and reduce the sensitivity to TMZ in GBM.

## 4. Discussion

GBM is the most common and malignant intracranial tumor in adults, and the current standard first-line chemotherapy for GBM is TMZ. Some GBM patients are resistant to TMZ, and these patients have a poor prognosis. The mechanism of TMZ resistance is very complex, and exploration of this mechanism is a hot research topic in this field. Studies have shown that EGFRvIII is associated with poor prognosis of GBM patients, but the reason for this is not clear. In this study, we found that the expression of EGFRvIII was closely related to GBM and could affect the gene expression profile of tumor tissues in patients with GBM. EGFRvIII was highly expressed in clinical tissue samples of GBM patients. In addition, GBM patients with EGFRvIII (−) were found to be sensitive to TMZ chemotherapy, while GBM patients with EGFRvIII (+) were found to be less sensitive to TMZ. These results fully indicate that EGFRvIII plays an important role in the chemotherapy resistance of TMZ in GBM, which is consistent with other reports [55,56,57]. To date, the reason for low sensitivity to TMZ in GBM patients with EGFRvIII (+) observed in this study has not been conclusively determined.

Epithelial–mesenchymal transition (EMT) is a common biological process undergone during the metastasis of a variety of malignant tumor cells [58,59]. A malignant process similar to EMT also occurs in the evolution of GBM, called proneural–mesenchymal transition. In both in vivo and in vitro studies, EGFRvIII (+) cells showed decreased expression of PN GBM markers and increased expression of MES GBM markers. Combined with our observation that EGFRvIII (+) cells exhibited the enhanced cell viability and migration ability, these results suggested that EGFRvIII (+) cells may have undergone a malignant transformation known as proneural–mesenchymal transition, which has not been reported to date as far as we are concerned.

In this study, we found that EGFRvIII (+) GBM cell lines and glioma stem cell lines showed low sensitivity to TMZ. Studies have reported that proneural–mesenchymal transition is one of the important causes of TMZ resistance [42], but the mechanism involved in GBM proneural–mesenchymal transition is extremely complex, including cytokines, transcription factors, and signaling pathways [11,12,47,49,60]. In addition, recent studies have confirmed that the amount of some metabolism-related enzymes in MES GBM is significantly higher than that in PN GBM, suggesting that the metabolic pathway of GBM may undergo some changes during its proneural–mesenchymal transition.

ALDH is an important metabolic enzyme involved in the occurrence and development of various tumors [24]. The expression of ALDH1A3 has significant effects on cell apoptosis, cell proliferation, glucose consumption, and lactic acid metabolism [61]. It has been reported that ALDH1A3 is closely related to phenotype in GBM, and the cells highly expressed by ALDH1A3 have stem cell characteristics [62,63]. In this study, the downstream target genes of EGFRvIII were screened using the second-generation transcriptome sequencing results. qRT-PCR and Western blot further confirmed that the mRNA and protein expression of ALDH1A3 were increased in GBM cell line and glioma stem cell line expression of EGFRvIII (+), indicating that ALDH1A3 was the downstream target gene of EGFRvIII. These results indicated that EGFRvIII might influence proneural–mesenchymal transition and TMZ sensitivity in GBM cells by regulating ALDH1A3 expression. Unfortunately, we do not know exactly which signaling pathway is involved in this process.

Studies have shown that the activation of the NF-κB signaling pathway is closely related to the resistance of chemotherapy drug TMZ [64,65]. In the current study, we found that GBM cells with EGFRvIII (+) increased the expression of phosphorylated NF-κB, which is consistent with previous reports. Canino et al. found the inhibition of the NF-κB signaling pathway affected the activity of the *ALDH1A3* promoter in the pleural mesothelioma. In this study, LASAGNA software and motif analysis were used to predict the binding of NF-κB to the *ALDH1A3* promoter region, and ChIP experiments confirmed the existence of binding of NF-κB to the *ALDH1A3* promoter region. These results fully indicate that NF-κB can transcriptionally regulate *ALDH1A3* in GBM cells. After further administration of the NF-κB inhibitor, the expression of ALDH1A3 in EGFRvIII (+) cells is decreased, suggesting that EGFRvIII (+) cells may trigger the binding of NF-κB to the *ALDH1A3* promoter by activating NF-κB. These results provide important theoretical and experimental basis for us to further understand the relationship between NF-κB and ALDH1A3. We also found that the inhibition of NF-κB could reverse the enhancement of cell migration caused by EGFRvIII, the decrease in the expression of PN marker (SOX2), the increase in MES marker (CD44), and the increase in the IC_50_ value of TMZ. These results together suggest that EGFRvIII can regulate the expression of ALDH1A3 by activating the NF-κB pathway to promote proneural–mesenchymal transition of GBM and reduce its sensitivity to TMZ, which may be one of the important reasons why GBM patients with EGFRvIII (+) have low sensitivity to TMZ, potentially providing a new idea for clinical research and development of GBM treatment drugs.

There are some shortcomings in this study. For example, we analyzed the signaling pathways associated with EGFRvIII downstream of the CGGA database and the second-generation transcriptome sequencing data of GBM cell lines and explored the NF-kB signaling pathway in depth. However, whether there are other important pathways involved in proneural–mesenchymal transition in EGFRvIII (+) GBM cells and the sensitivity to TMZ need to be further investigated.

In conclusion, we found that EGFRvIII expression is closely related to GBM and can affect the gene expression profile of tumor tissue in GBM patients; GBM patients with EGFRvIII (+) have low sensitivity to TMZ; and EGFRvIII regulates the expression of ALDH1A3 by activating NF-κB, thus promoting the proneural–mesenchymal transition of GBM and reducing its sensitivity to TMZ. These results will provide experimental evidence for the clinical selection of GBM patients with EGFRvIII (+).

## 5. Conclusions

Our findings suggest that EGFRvIII promotes the proneural–mesenchymal transition of GBM and reduces its sensitivity to TMZ. In addition, our results suggest that EGFRvIII activates the NF-κB signaling pathway and regulates of the expression of ALDH1A3. Conclusively, our findings demonstrate that EGFRvIII promotes the proneural–mesenchymal transition of GBM and reduces its sensitivity to TMZ by regulating the NF-κB/ALDH1A3 axis, which providing an experimental basis for the selection of clinical drugs for GBM patients with EGFRvIII (+).

## Figures and Tables

**Figure 1 genes-14-00651-f001:**
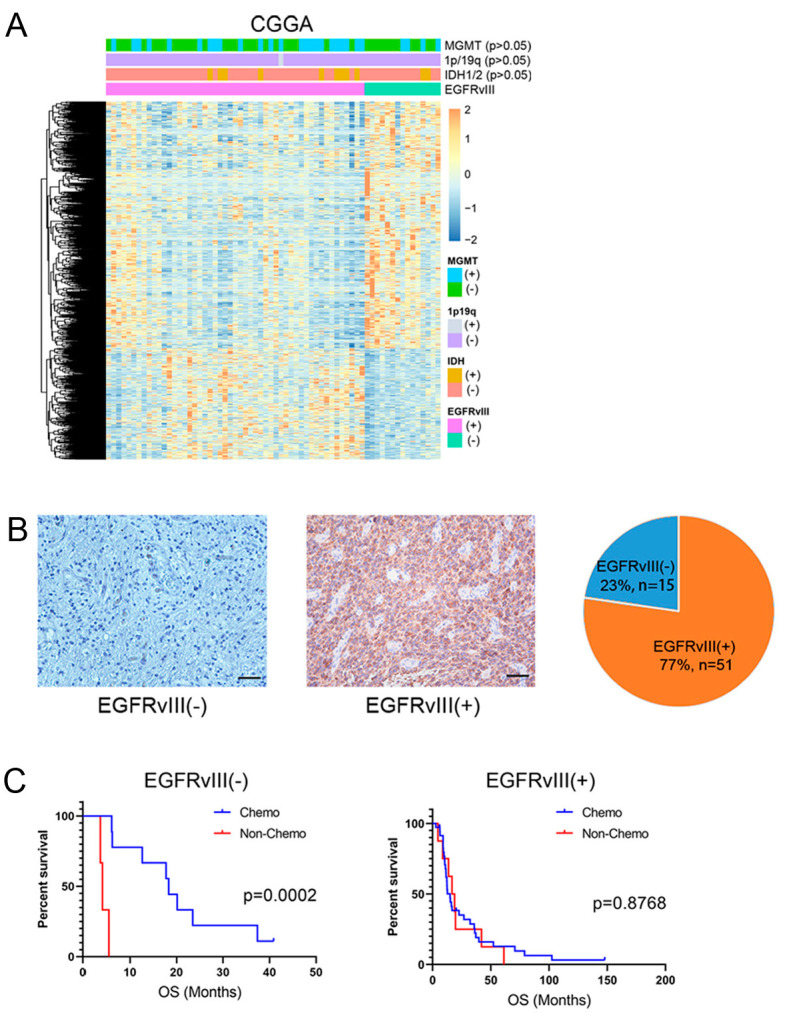
EGFRvIII affected the gene expression profile of GBM patients, and GBM patients with EGFRvIII (+) had low sensitivity to TMZ. (**A**) Analysis of the differences of gene expression profiles in tumor tissues of GBM patients with EGFRvIII (−) or EGFRvIII (+) in the CGGA database. (**B**) Immunohistochemical staining of EGFRvIII in tumor tissues of GBM patients and statistics of EGFRvIII expression in 66 GBM patients (scale bars: 50 μm). (**C**) Effect of EGFRvIII expression on overall survival of GBM patients with or without TMZ. *p* = 0.0002 and *p* = 0.8768 by Kaplan–Meier tests.

**Figure 2 genes-14-00651-f002:**
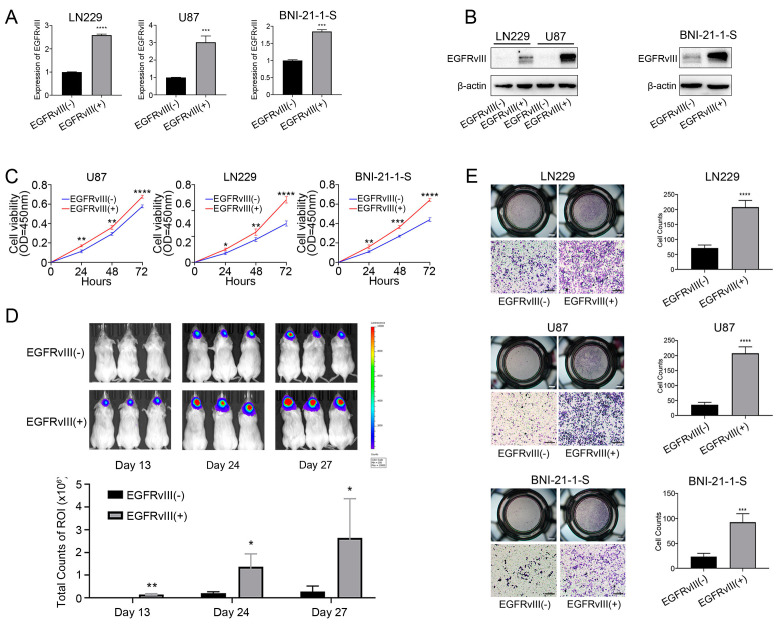
GBM cells and glioma stem cells with EGFRvIII (+) significantly increased cell viability and cell migration. (**A**) The mRNA expression of EGFRvIII was detected by qRT-PCR in U87-EGFRvIII (+) cells, LN229-EGFRvIII (+) cells, and BNI-21-1-S-EGFRvIII (+) cells. (**B**) The protein expression of EGFRvIII was detected by Western blot. (**C**) The cell proliferation was detected by CCK-8 in U87-EGFRvIII (+) cells, LN229-EGFRvIII (+) cells, and BNI-21-1-S-EGFRvIII (+) cells. (**D**) The tumor growth of GBM in the orthotopic xenograft tumor model of mice was detected by in vivo bioluminescence imaging (*n* = 10 per group). (**E**) The cell migration was detected by Transwell in U87-EGFRvIII (+) cells, LN229-EGFRvIII (+) cells, and BNI-21-1-S-EGFRvIII (+) cells (scale bar: 100 μm). Representative data are determined from three independent experiments (*n* = 3 per group) vs. EGFRvIII (−). * *p* < 0.05, ** *p* < 0.01, *** *p* < 0.001, **** *p* < 0.0001 by Student’s two-tailed *t*-test.

**Figure 3 genes-14-00651-f003:**
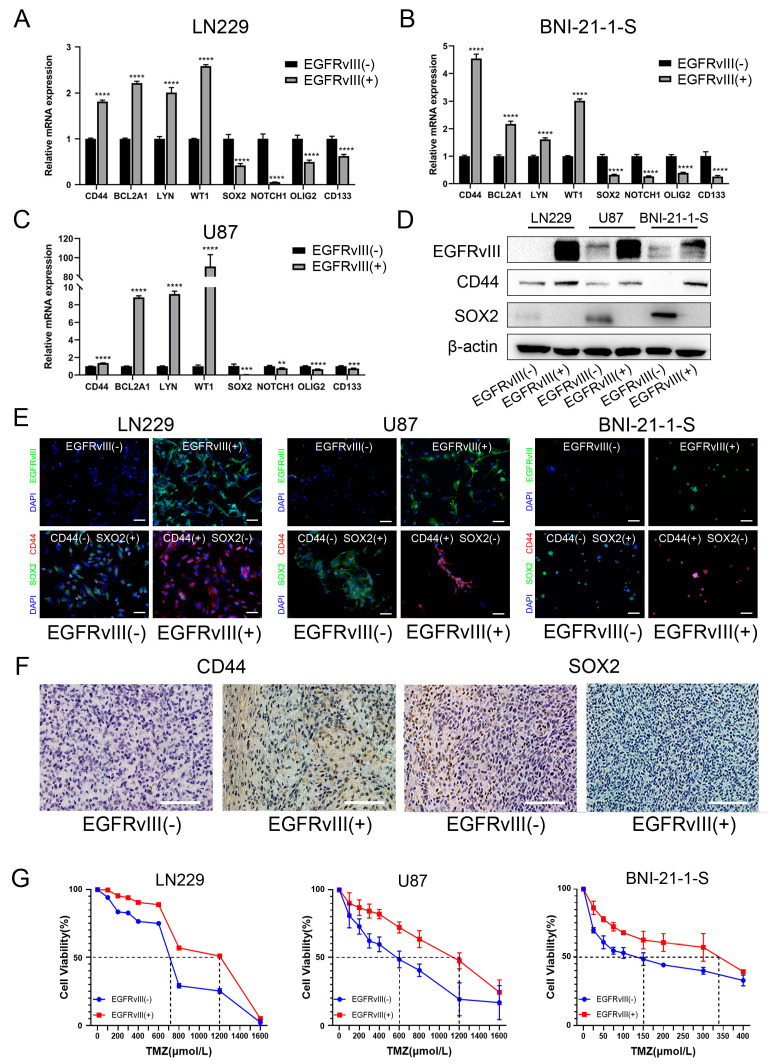
GBM cells and glioma stem cells with EGFRvIII (+) significantly decreased the expression of PN GBM markers, increased the expression of MES GBM markers, and increased the IC_50_ value of TMZ. (**A**–**C**) The mRNA expression of PMT-related markers were detected by qRT-PCR in U87-EGFRvIII (+) cells, LN229-EGFRvIII (+) cells, and BNI-21-1-S-EGFRvIII (+) cells. (**D**) The protein expressions of PMT-related markers were detected by Western blot. (**E**) The expression of PMT-related markers detected by immunofluorescence (green for EGFRvIII and SOX2, red for CD44, blue for DAPI) (scale bars: 50 μm). (**F**) The expressions of PMT-related markers were detected by immunohistochemistry in tumor tissue from the orthotopic xenograft tumor model of mice (*n* = 3 per group) (scale bars: 100 μm). (**G**) The half-maximal inhibitory concentration of U87-EGFRvIII (+) cells, LN229-EGFRvIII (+) cells, and BNI-21-1-S-EGFRvIII (+) cells on TMZ was detected by CCK8. Representative data are determined from three independent experiments (*n* = 3 per group) vs. EGFRvIII (−). ** *p* < 0.01, *** *p* < 0.001, **** *p* < 0.0001 by two-tailed Student’s *t*-test.

**Figure 4 genes-14-00651-f004:**
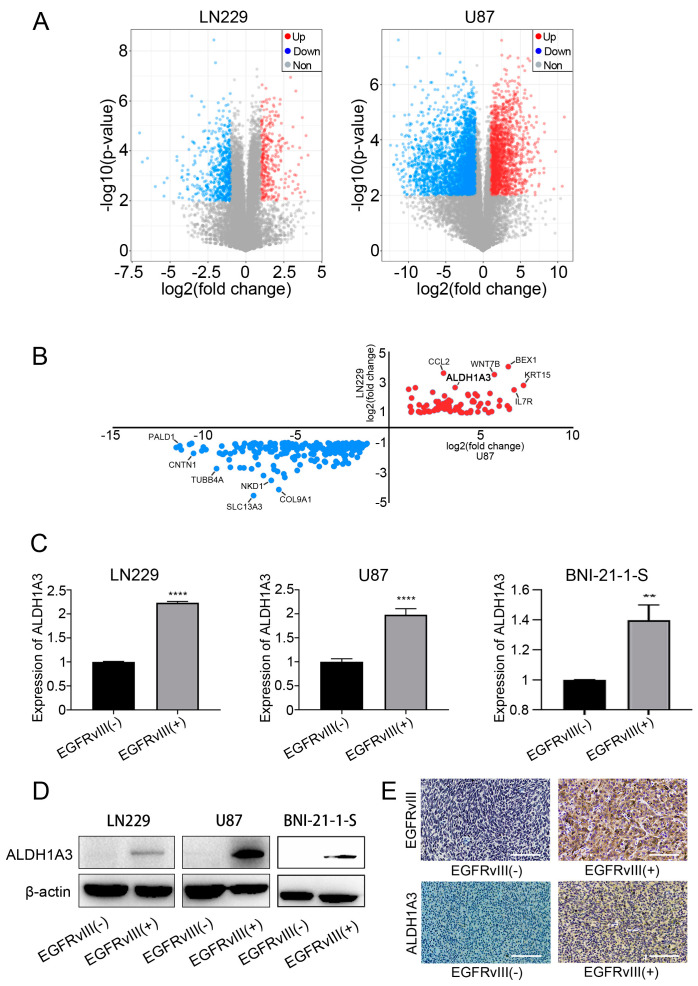
*ALDH1A3* was screened and identified as the downstream target gene of EGFRvIII. (**A**) Differential gene expression analysis of LN229-EGFRvIII (+) cells and U87-EGFRvIII (+) cells. (**B**) Analysis of common differential gene expression of U87-EGFRvIII (+) cells and LN229-EGFRvIII (+) cells. (**C**) The mRNA expression of *ALDH1A3* was detected by qRT-PCR in U87-EGFRvIII (+) cells, LN229-EGFRvIII (+) cells, and BNI-21-1-S-EGFRvIII (+) cells. (**D**) The protein expression of ALDH1A3 was detected by Western blot. (**E**) The expression of EGFRvIII and ALDH1A3 was detected by immunohistochemistry in tumor tissue from the orthotopic xenograft tumor model of mice (scale bars: 100 μm) (*n* = 3 per group). Representative data are determined from three independent experiments (*n* = 3 per group). vs. EGFRvIII (−). ** *p* <0.01, **** *p* <0.0001 by the two-tailed Student’s *t*-test.

**Figure 5 genes-14-00651-f005:**
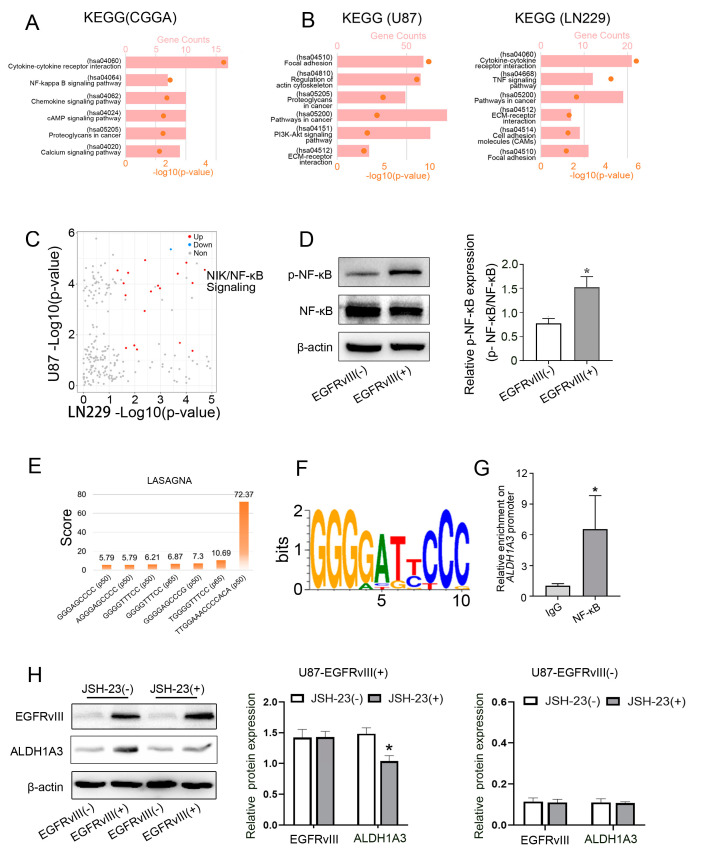
EGFRvIII regulated ALDH1A3 expression through the NF-κB signaling pathway. (**A**) Screening of the EGFRvIII-related signaling pathway in the CGGA database by KEGG analysis. (**B**) KEGG analysis of U87-EGFRvIII (+) cells and LN229-EGFRvIII (+) cells. (**C**) GSVA analysis of U87-EGFRvIII (+) cells and LN229-EGFRvIII (+) cells. (**D**) The expression of p-NF-κB was detected by Western blot in U87-EGFRvIII (+) cells. (**E**) LASAGNA analysis. (**F**) Motif analysis. (**G**) NF-κB binding to *ALDH1A3* gene promoter region detected by the ChIP experiment in U87-EGFRvIII (+) cells. (**H**) The expression of EGFRvIII and ALDH1A3 was detected by Western blot in U87-EGFRvIII (−) cell and U87-EGFRvIII (+) cell pretreatment with or without an NF-κB inhibitor (JSH-23), and statistical analysis of the expression of EGFRvIII and ALDH1A3. Representative data are determined from three independent experiments (*n* = 3 per group) vs. EGFRvIII (−). * *p* < 0.05 by the two-tailed Student’s *t*-test.

**Figure 6 genes-14-00651-f006:**
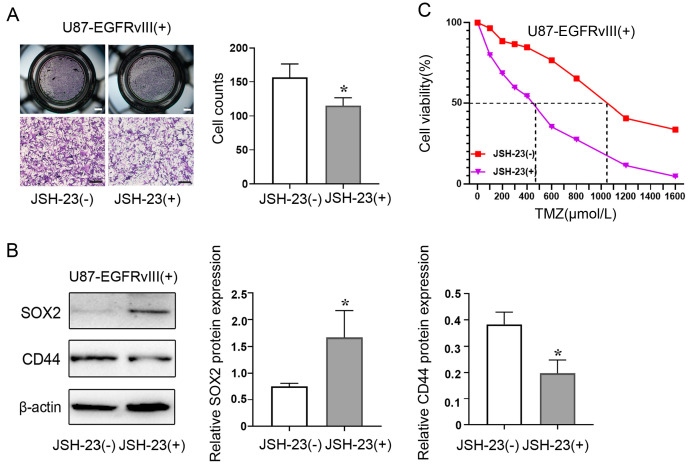
EGFRvIII affected the migration, PMT of U87-EGFRvIII (+) cells, and TMZ sensitivity of U87-EGFRvIII (+) cells by regulating the NF-κB signaling pathway. (**A**) The cell migration was detected by Transwell assay in U87-EGFRvIII (+) cell pretreatment with NF-κB inhibitor (JSH-23) (scale bar: 100 μm). (**B**) The protein expression of CD44 and SOX2 was detected by Western blot in U87-EGFRvIII (+) cell pretreatment with NF-κB inhibitor (JSH-23). Representative data are determined from three independent experiments (*n* = 3 per group) vs. JSH-23 (−). * *p* < 0.05 by two-tailed Student’s *t*-test. (**C**) The half maximal inhibitory concentration of U87-EGFRvIII (+) cell pretreatment with NF-κB inhibitor (JSH-23) was detected by CCK8 assay.

**Table 1 genes-14-00651-t001:** Correlation between EGFRvIII expression and clinical data in 66 patients with glioblastoma multiforme.

Characteristics	EGFRvIII (−) *n* = 15	EGFRvIII (+) *n* = 51	Variation (*p*-Value)
Age at diagnosis			0.2136 ^a^
Mean	44.13	48.92	
Standard Deviation	15.50	12.18	
Gender			0.7654 ^b^
Male	10	30	
Female	5	21	
IDH1 Mutation			0.9999 ^b^
Mutation	2	9	
Wildtype	13	42	
Not Available			
1p/19q Codeletion Status			0.9999 ^b^
Codeletion	0	0	
Non-Codeletion	15	50	
Not Available	0	1	
*MGMT* Promoter Status			0.2361 ^b^
Methylated	4	24	
Un-Methylated	11	27	
Not Available			
Radiotherapy			0.4200 ^b^
Yes	12	44	
No	3	6	
Not Available	0	1	
Chemotherapy			0.2825 ^b^
Yes	10	41	
No	5	9	
Not Available	0	1	

^a^ Unpaired t test; ^b^ Fisher’s exact test.

## Data Availability

Not acceptable.

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
