# Peer review of "EGFRvIII Promotes the Proneural–Mesenchymal Transition of Glioblastoma Multiforme and Reduces Its Sensitivity to Temozolomide by Regulating the NF-κB/ALDH1A3 Axis"

_genes, 2023, doi:10.3390/genes14030651_

Round 1

Reviewer 1 Report

In their manuscript, Shi et al., reports that EGFRvIII expression underlies insensitivity of GBM via promoting PN->MES (proneural to mesenchymal) transition of tumor cells.  This transition is shown to be accomplished via activation of the transcription factor NF-kB, which, in turn, stimulates the expression of ALDH1A3, which is known to have effects on cell proliferation, glucose conversion and lactic acid metabolism.  It is also known that GBM cells expressing high levels of ALDH1A3 have stem cell like characteristics. Thus, their experimental approach, results obtained, and conclusions appear quite reasonable.

The authors arrive at their conclusions as follows:

1.       They analyzed gene expression profiles of GBM patients and find that 77% of the patients express EGFRvIII, and that the overall gene expression profiles between EGFRvIII+ and EGFRvIII- tumors is significantly different.  They also find that overall survival of EGFRvIII+ patients was independent of TMZ.

2.       They constructed EGFRvIII+ GBM and glioma stem cell lines and show that these EGFRvIII+ cells had significantly increased viability and migration potential.

3.       By employing qRT-PCR, WB and HIS, they observe that EGFRvIII+ glioma and glioma stem cell lines had decreased levels of proneural (PN) GBM markers and elevated expression of mesenchymal (MES) GBM markers.  They also show that the IC50 values of cells expressing EGFRvIII were significantly higher than cells not expressing the mutant receptor.  These results imply that EGFRvIII has a role in reducing the sensitivity of GBM and GBM stem cells to TMZ.

4.       By employing transcriptome sequencing, qRT-PCR, WB and IHC, they show that ALDH1A3 expression is significantly higher in EGFRvIII+ cells compared to control cells. These results enable them to imply that ALDH1A3 is a downstream target gene of EGFRvIII.

5.       Analyzing signaling pathway and by employing GSVA analysis of transcript sequencing of brain tumor tissues and GBM cell lines, they infer that a pathway involving NF-kB is associated with EGFRvIII.  Further, using ChiP assay they show that NF-kB is bound to the promoter region of ALDH1A3.  In addition, they show that inhibiting NF-kB expression also decreases expression of ALDH1A3 without affecting expression level of EGFRvIII.  From these results, they propose that NF-kB transcriptionally regulates ALDH1A3 expression, which is induced by EGFRvIII.

6.       Finally, using WB and other means, they show that inhibiting NF-kB expression by JS-23 decreased expression of ALDH1A3, affected migration of EGFRvIII+ cells, decreased PN GBM marker (SOX2) and increased MES GBM marker (CD44) in these cells.

I feel the successive experiments performed are logical and appropriate to explore the underlying issue and the demonstration of a potential NF-kB-AlDH1A3 axis in EGFRvIII+ GBM cells appears to be novel and important and could have clinical implication.

Some minor issues:

1.       Briefly discuss why only 50% of the signaling pathways shown are common between U87 EGFRvIII+ and LN-229 EGFRvIII+ cells?  Also comment on why none of the signaling pathways shown is common between the three sets.

2.       Data presented in Fig 4B indicate there are other genes besides ALDH1A3 that may be as relevant as ALDH1A3.  Briefly explain why the other genes were also not evaluated in later experiments.

3.       I feel that the first paragraph and about half of the second paragraph in the Discussion is suitable for Introduction not Discussion. Consequently, the Discussion needs to be re-written.

Author Response

Dear Reviewer:

Thank you for your comments concerning our manuscript entitled “EGFRvIII Promotes the Proneural-Mesenchymal Transition of Glioblastoma Multiforme and Reduces Its Sensitivity to Temozolomide by Regulating NF-κB/ALDH1A3 Axis”, (ID: genes-2203856).

These comments concerning our manuscript are all valuable and very helpful for revising and improving our paper, and have the important guiding significance for our future research. We have tried our best to revise our manuscript according to the comments. These revisions will not influence the content and framework of the paper. Revised parts are marked in yellow in the revised version. The main corrections in the paper and the responds to the reviewer's comments are as flowing:

Responds to the reviewer's comments:

Reviewer 1:

In their manuscript, Shi et al., reports that EGFRvIII expression underlies insensitivity of GBM via promoting PN->MES (proneural to mesenchymal) transition of tumor cells.  This transition is shown to be accomplished via activation of the transcription factor NF-kB, which, in turn, stimulates the expression of ALDH1A3, which is known to have effects on cell proliferation, glucose conversion and lactic acid metabolism.  It is also known that GBM cells expressing high levels of ALDH1A3 have stem cell like characteristics. Thus, their experimental approach, results obtained, and conclusions appear quite reasonable.

The authors arrive at their conclusions as follows:

  1. They analyzed gene expression profiles of GBM patients and find that 77% of the patients express EGFRvIII, and that the overall gene expression profiles between EGFRvIII+ and EGFRvIII- tumors is significantly different.  They also find that overall survival of EGFRvIII+ patients was independent of TMZ.
  2. They constructed EGFRvIII+ GBM and glioma stem cell lines and show that these EGFRvIII+ cells had significantly increased viability and migration potential.
  3. By employing qRT-PCR, WB and HIS, they observe that EGFRvIII+ glioma and glioma stem cell lines had decreased levels of proneural (PN) GBM markers and elevated expression of mesenchymal (MES) GBM markers.  They also show that the IC50 values of cells expressing EGFRvIII were significantly higher than cells not expressing the mutant receptor.  These results imply that EGFRvIII has a role in reducing the sensitivity of GBM and GBM stem cells to TMZ.
  4. By employing transcriptome sequencing, qRT-PCR, WB and IHC, they show that ALDH1A3 expression is significantly higher in EGFRvIII+ cells compared to control cells. These results enable them to imply that ALDH1A3 is a downstream target gene of EGFRvIII.
  5. Analyzing signaling pathway and by employing GSVA analysis of transcript sequencing of brain tumor tissues and GBM cell lines, they infer that a pathway involving NF-kB is associated with EGFRvIII.  Further, using ChiP assay they show that NF-kB is bound to the promoter region of ALDH1A3.  In addition, they show that inhibiting NF-kB expression also decreases expression of ALDH1A3 without affecting expression level of EGFRvIII.  From these results, they propose that NF-kB transcriptionally regulates ALDH1A3 expression, which is induced by EGFRvIII.
  6. Finally, using WB and other means, they show that inhibiting NF-kB expression by JS-23 decreased expression of ALDH1A3, affected migration of EGFRvIII+ cells, decreased PN GBM marker (SOX2) and increased MES GBM marker (CD44) in these cells.

I feel the successive experiments performed are logical and appropriate to explore the underlying issue and the demonstration of a potential NF-kB-AlDH1A3 axis in EGFRvIII+ GBM cells appears to be novel and important and could have clinical implication.

Some minor issues:

  1. Briefly discuss why only 50% of the signaling pathways shown are common between U87 EGFRvIII+ and LN-229 EGFRvIII+ cells?  Also comment on why none of the signaling pathways shown is common between the three sets.

Thank you for raising these questions.

In the current study, the results showed that 50% of the signaling pathways are common between U87 EGFRvIII (+) and LN-229 EGFRvIII (+) cells, and none of the signaling pathways is common between the three sets. Many factors including signaling pathways or cytokines are activated in different cell lines after transfection with EGFRvIII. However, these factors that play a major role in different cell lines are differences. The results only partially showed the first six factors that showed the significant effects of EGFRvIII on glioblastoma tissues and different cell lines.

  1. Data presented in Fig 4B indicate there are other genes besides ALDH1A3 that may be as relevant as ALDH1A3.  Briefly explain why the other genes were also not evaluated in later experiments.

Thank you for raising the question.

Just as you said that the data presented in Fig 4B indicate there are other genes besides ALDH1A3 that may be as relevant as ALDH1A3, studies have shown that ALDH1A3 plays an important role in the proneural-mesenchymal transition and malignant progression of GBM cells [1,2,3]. Our previous studies also showed that ALDH1A3 is one of the key metabolic enzymes involved in the occurrence and development of GBM [4,5,6]. It is suggested that ALDH1A3 may be the target gene downstream of EGFRvIII. Therefore, in the current study, we focused on the important role and mechanism of ALDH1A3 by which EGFRvIII promoted the proneural-mesenchymal transition of GBM and reduces GBM sensitivity to temozolomide, which has not been reported so far. In future studies, we will further explore the role and the related mechanisms of other genes in this process.

References:

  • Li J, Garavaglia S, Ye Z, Moretti A, Belyaeva OV, Beiser A, Ibrahim M, Wilk A, McClellan S, Klyuyeva AV, Goggans KR, Kedishvili NY, Salter EA, Wierzbicki A, Migaud ME, Mullett SJ, Yates NA, Camacho CJ, Rizzi M, Sobol RW. A specific inhibitor of ALDH1A3 regulates retinoic acid biosynthesis in glioma stem cells. Commun Biol. 2021 Dec 21;4(1):1420. doi: 10.1038/s42003-021-02949-7. PMID: 34934174; PMCID: PMC8692581.
  • Poturnajova M, Kozovska Z, Matuskova M. Aldehyde dehydrogenase 1A1 and 1A3 isoforms - mechanism of activation and regulation in cancer. Cell Signal. 2021 Nov;87:110120. doi: 10.1016/j.cellsig.2021.110120. Epub 2021 Aug 21. PMID: 34428540; PMCID: PMC8505796.
  • Zirjacks L, Stransky N, Klumpp L, Prause L, Eckert F, Zips D, Schleicher S, Handgretinger R, Huber SM, Ganser K. Repurposing Disulfiram for Targeting of Glioblastoma Stem Cells: An In Vitro Study. Biomolecules. 2021 Oct 21;11(11):1561. doi: 10.3390/biom11111561. PMID: 34827559; PMCID: PMC8615869.
  • Li G, Li Y, Liu X, Wang Z, Zhang C, Wu F, Jiang H, Zhang W, Bao Z, Wang Y, Cai J, Zhao L, Kahlert UD, Jiang T, Zhang W. ALDH1A3 induces mesenchymal differentiation and serves as a predictor for survival in glioblastoma. Cell Death Dis. 2018 Dec 11;9(12):1190. doi: 10.1038/s41419-018-1232-3. PMID: 30538217; PMCID: PMC6290011.
  • Zhang W, Liu Y, Hu H, Huang H, Bao Z, Yang P, Wang Y, You G, Yan W, Jiang T, Wang J, Zhang W. ALDH1A3: A Marker of Mesenchymal Phenotype in Gliomas Associated with Cell Invasion. PLoS One. 2015 Nov 17;10(11):e0142856. doi: 10.1371/journal.pone.0142856. PMID: 26575197; PMCID: PMC4648511.
  • Zhang W, Yan W, You G, Bao Z, Wang Y, Liu Y, You Y, Jiang T. Genome-wide DNA methylation profiling identifies ALDH1A3 promoter methylation as a prognostic predictor in G-CIMP- primary glioblastoma. Cancer Lett. 2013 Jan 1;328(1):120-5. doi: 10.1016/j.canlet.2012.08.033. Epub 2012 Sep 5. PMID: 22960273.

  1. I feel that the first paragraph and about half of the second paragraph in the Discussion is suitable for Introduction not Discussion. Consequently, the Discussion needs to be re-written.

Thank you for raising the question.

According to your suggestion, we have moved the content of the first paragraph and about half of the second paragraph in the Discussion to the Introduction, and re-written the Introduction and Discussion (marked in yellow).

Once again, thank you very much for your comments and suggestions.

Sincerely yours,

Zhongfang Shi

*Corresponding authors: Xiaoguang Yu, Department of Biochemistry & Molecular Biology, Harbin Medical University, 157 Baojian Road, Harbin 150081, Heilongjiang, China. Email: [email protected].

Reviewer 2 Report

First of all, the current glioblastoma molecular subtypes are only proneural, mesenchymal and classical. The neural subtype has been considered as a "noise" or residual normals cells from specimens in the article of Wang et al. 2017. Additionally, in sentence 42 the authors talk about metastasis and I suggest deleting metastasis from the text, because glioblastoma, usually, does not present metastasis. Finally, in the introduction section, the authors talk about their previous clinical study (sentences 52-56), but there is no reference. Has the study been published? I suggest adding the reference if it is available.

From who is the BNI-21-1-S glioblastoma cell line? How was obtained? I suggest adding a reference to the first study where the cell line has been used or the group from which it comes.

During the methods section, not all products are well referenced; some references, catalogue numbers, the country of products, etc. Are missing. Please, review that all references are correctly added.

All in vivo analyses were performed with female mice, why?

I suggest deleting sentences 210-212 from the methodology section, due it is a conclusion or result.

Which genes of EGFRvIII or cell cluster (sentence 221) were used in David's analysis? I suggest adding a supplementary table with the genes to corroborate their role in the cell cluster or their relationship with EGFR and make the study more reproductive.

The statistical analysis is not complete. There is missing the fisher test for the non-parametric test obtained in table 1, and the t-test for two groups comparison. On which occasion, was the ANOVA test performed? In which analysis multi-group comparison was performed? I suggest adding the test information in the legend of the figures.

In addition, the CGGA analysis and LASAGNA analysis information is missing in the methods section. I suggest adding the section for this analysis with the reference and the parameters that the authors used to perform the analysis.

During the CGGA analysis, the authors comment that there are no differences between the EGFRvIII group and the molecular typing markers but then comment that the different expression of EGFRvIII may lead to the different biological characteristics of GBM cells, why conclude this?

Is the classification of the EGFRvIII group on the CGGA database dual, ergo only positive or negative, or there is information about the EGFRvIII expression?

I think the reference to selecting the PN and MES subtypes is not the most appropriate. Could you corroborate that CD133 is only expressed in the PN subtype? CD133 is a stem cell marker and might be expressed in the MES subtype too. Could the authors confirm that these genes are in the Verhaak or Wang signature?

Which number of replicates were used in each experiment? I suggest adding this information to confirm that the statistical analysis was correctly applied and the number of experiments is sufficient to obtain statistical significance. If there are fewer than three replicates and two cell lines used in any experiment, the authors should to performed more experiments to make the final conclusions.

In addition, why the authors used only the two GBM cell lines and not the GSC for ALDH1A3 analysis? Has studied this topic in the third GBM cell line? In addition, has the authors studied the expression of ALDH1A3 in the CGGA database or human specimens? Are any relation in its expression with EGFRvIII expression?

Which is the contribution of KEGG analysis to the study? Only in the KEGG analysis with the CGGA database, the pathways related to EGFRvIII are expressed. Is there another pathway related to PMT, for example, in the results of KEGG analysis with GBM cell lines? Are the authors comparing the KEEG analysis of the GBM cell line with EGFRvIII (+) Vs (-)? This could be more informative to compare the pathways activated with the overexpression of EGFRvIII.

To confirm that the expression of EGFRvIII is involved in the NFkB more experiments and more cell lines must be used. How many experiments were used to obtain figure 5D-H?

I think that in the legend of figure 6, there is some comment from some author or reviewer (sentences 485-387).

Finally, in the discussion section, E2F6 appears for the first time and there is not any related to this topic during the study. Is it important? Only there is this sentence, that's why I suggest deleting the sentence

Author Response

Dear Reviewer:

Thank you for your comments concerning our manuscript entitled “EGFRvIII Promotes the Proneural-Mesenchymal Transition of Glioblastoma Multiforme and Reduces Its Sensitivity to Temozolomide by Regulating NF-κB/ALDH1A3 Axis”, (ID: genes-2203856).

These comments concerning our manuscript are all valuable and very helpful for revising and improving our paper, and have the important guiding significance for our future research. We have tried our best to revise our manuscript according to the comments. These revisions will not influence the content and framework of the paper. Revised parts are marked in yellow in the revised version. The main corrections in the paper and the responds to the reviewer's comments are as flowing:

Responds to the reviewer's comments:

1 First of all, the current glioblastoma molecular subtypes are only proneural, mesenchymal and classical. The neural subtype has been considered as a "noise" or residual normals cells from specimens in the article of Wang et al. 2017. Additionally, in sentence 42 the authors talk about metastasis and I suggest deleting metastasis from the text, because glioblastoma, usually, does not present metastasis. Finally, in the introduction section, the authors talk about their previous clinical study (sentences 52-56), but there is no reference. Has the study been published? I suggest adding the reference if it is available.

Thank you for raising these questions.

Firstly, thank you for your suggestion. The genome wide transcriptome analyses suggest that GBM can be divided into four clinically relevant subtypes: proneural (PN), neural, classic, and mesenchymal (Mes) GBM [1,2]. Just like you said, recently, a revised TCGA classification system was reported for IDH-wild type GBM, and the transcriptional glioma subtypes defined through clustering based on tumor-intrinsic genes strongly overlapped with the proneural, classical, and mesenchymal subtypes but the neural subtype was considered to represent normal cell contamination [3]. We thought that two different classification methods for GBM have their own characteristics. In my manuscript, we referred to the four glioblastoma molecular subtypes based on gene expression analysis: classical, proneu­ral, mesenchymal and neural subtypes. According to your suggestion, we added to the description of Verhaak’s latest update for reclassifying GBM, and GBMs can be classified into three subtypes: proneural, classical, and mesenchymal subtypes based on transcriptional signatures (marked in yellow).

  • Phillips HS, Kharbanda S, Chen R, Forrest WF, Soriano RH, Wu TD, Misra A, Nigro JM, Colman H, Soroceanu L, Williams PM, Modrusan Z, Feuerstein BG, Aldape K. Molecular subclasses of high-grade glioma predict prognosis, delineate a pattern of disease progression, and resemble stages in neurogenesis. Cancer Cell. 2006 Mar;9(3):157-73. doi: 10.1016/j.ccr.2006.02.019. PMID: 16530701.
  • Verhaak RG, Hoadley KA, Purdom E, Wang V, Qi Y, Wilkerson MD, Miller CR, Ding L, Golub T, Mesirov JP, Alexe G, Lawrence M, O'Kelly M, Tamayo P, Weir BA, Gabriel S, Winckler W, Gupta S, Jakkula L, Feiler HS, Hodgson JG, James CD, Sarkaria JN, Brennan C, Kahn A, Spellman PT, Wilson RK, Speed TP, Gray JW, Meyerson M, Getz G, Perou CM, Hayes DN; Cancer Genome Atlas Research Network. Integrated genomic analysis identifies clinically relevant subtypes of glioblastoma characterized by abnormalities in PDGFRA, IDH1, EGFR, and NF1. Cancer Cell. 2010 Jan 19;17(1):98-110. doi: 10.1016/j.ccr.2009.12.020. PMID: 20129251; PMCID: PMC2818769.
  • Wang Q, Hu B, Hu X, Kim H, Squatrito M, Scarpace L, deCarvalho AC, Lyu S, Li P, Li Y, Barthel F, Cho HJ, Lin YH, Satani N, Martinez-Ledesma E, Zheng S, Chang E, Sauvé CG, Olar A, Lan ZD, Finocchiaro G, Phillips JJ, Berger MS, Gabrusiewicz KR, Wang G, Eskilsson E, Hu J, Mikkelsen T, DePinho RA, Muller F, Heimberger AB, Sulman EP, Nam DH, Verhaak RGW. Tumor Evolution of Glioma-Intrinsic Gene Expression Subtypes Associates with Immunological Changes in the Microenvironment. Cancer Cell. 2017 Jul 10;32(1):42-56.e6. doi: 10.1016/j.ccell.2017.06.003. Erratum in: Cancer Cell. 2018 Jan 8;33(1):152. PMID: 28697342; PMCID: PMC5599156.

Secondly, as you said, studies have reported that the extracranial metastasis of GBM is extremely rare, occurring in <2% of all GBMs [1,2]. We have deleted metastasis from the text.

  • Ray A, Manjila S, Hdeib AM, Radhakrishnan A, Nock CJ, Cohen ML, Sloan AE. Extracranial metastasis of gliobastoma: Three illustrative cases and current review of the molecular pathology and management strategies. Mol Clin Oncol. 2015 May;3(3):479-486. doi: 10.3892/mco.2015.494. Epub 2015 Jan 23. PMID: 26137254; PMCID: PMC4471516.
  • Piccirilli M, Brunetto GM, Rocchi G, Giangaspero F, Salvati M. Extra central nervous system metastases from cerebral glioblastoma multiforme in elderly patients. Clinico-pathological remarks on our series of seven cases and critical review of the literature. Tumori. 2008 Jan-Feb;94(1):40-51. doi: 10.1177/030089160809400109. PMID: 18468334.

Lastly, thank you for your suggestion. We are sorry for not description of our experimental results clearly. According to your suggestion, we have revised the description (marked in yellow).

2 From who is the BNI-21-1-S glioblastoma cell line? How was obtained? I suggest adding a reference to the first study where the cell line has been used or the group from which it comes.

Thank you for raising the question.

The BNI-21-1-S human glioma stem cell lines were acquired from patient specimens at Beijing Tiantan Hospital under protocols approved by the Institutional Review Board (IRB), and maintained as tumorspheres in GSC media. We have added the description in the methods section (marked in yellow).

3 During the methods section, not all products are well referenced; some references, catalogue numbers, the country of products, etc. Are missing. Please, review that all references are correctly added.

Thank you for your suggestion.

We are sorry for not description of all products clearly. According to your suggestion, we have added the information of the products in the methods section (marked in yellow).

4 All in vivo analyses were performed with female mice, why?

Thank you for raising the question.

Female mice are generally docile and do not fight; however, male mice kept in groups often fight, and keeping two or more male mice together often does not work well, which can seriously affect the establishment of experimental models.

5 I suggest deleting sentences 210-212 from the methodology section, due it is a conclusion or result.

Thank you for your suggestion.

We are sorry for the negligence. We have deleted sentences 210-212 from the methodology section.

6 Which genes of EGFRvIII or cell cluster (sentence 221) were used in David's analysis? I suggest adding a supplementary table with the genes to corroborate their role in the cell cluster or their relationship with EGFR and make the study more reproductive.

Thank you for raising the question.

We are sorry for not description of the genes of EGFRvIII used in David's analysis.

According to your suggestion, we have added a supplementary table with the genes to corroborate their relationship with EGFRvIII.

7 The statistical analysis is not complete. There is missing the fisher test for the non-parametric test obtained in table 1, and the t-test for two groups comparison. On which occasion, was the ANOVA test performed? In which analysis multi-group comparison was performed? I suggest adding the test information in the legend of the figures.

Thank you for raising the question.

We are sorry for the incomplete statistical analysis, and we also found that it was not appropriate that one-way ANOVA test was used in this experiment. Since what we compared are two unpaired groups and the data are normally distributed, we used Student's t test for analysis. According to your suggestion, we have added the test information in statistical analysis section and the legend of the figures (marked in yellow).

All statistical computations were performed with the statistical software environment R (version 4.2.1), SPSS statistical package (version 19), GraphPad Prism (version 9). All data were expressed as mean ± SD. Student's t test was used to compare differences between two groups. The Kaplan–Meier method with a two-sided log-rank test was used to compare the overall survival of patients between different groups. Unpaired t test and Fisher's exact test were used to analysis the correlation between EGFRvIII expression and clinical data in 66 patients with GBM. All experiments were repeated for three times. A p-value of <.05 was considered statistically significant.

8 In addition, the CGGA analysis and LASAGNA analysis information is missing in the methods section. I suggest adding the section for this analysis with the reference and the parameters that the authors used to perform the analysis.

Thank you for your suggestion.

We are sorry for not description of the CGGA analysis and LASAGNA analysis information in the methods section.

According to your suggestion, we have added the section for this analysis with the reference and the parameters used to perform the analysis (marked in yellow).

Transcriptome sequencing data and most molecular pathological information (MGMT status, Chromosome 1p/19q status, IDH status, and transcription subtypes) of glioma patients were obtained from CGGA official website (www.cgga.org.cn). The EGFRvIII status was obtained from the immunohistochemical staining of the above clinical samples. Bioinformatics analysis was performed through R 4.2.1 and GraphPad Prism 9 software.

Transcription factors were predicted using LASAGNA-Search 2.0 online tool (https://biogrid-lasagna.engr.uconn.edu/lasagna_search/). p50 and p65 were selected as TF Model Input, and promoter region (-1000 to 0 relative to TSS) of ALDH1A3 were selected as Promoter Sequence Input [1].

9 During the CGGA analysis, the authors comment that there are no differences between the EGFRvIII group and the molecular typing markers but then comment that the different expression of EGFRvIII may lead to the different biological characteristics of GBM cells, why conclude this?

Thank you for raising the question.

We are sorry for not description of our experimental results clearly. We have revised the description (marked in yellow).

The results showed that there were significant differences in the gene expression profile of tumor tissue between the two groups of GBM patients with EGFRvIII (-) and (+) (p <0.05), and no significant difference in the distribution of important molecular typing markers (including IDH1/2 status, 1p/19q codeletion status, MGMT promoter methyla-tion state) were found between the two groups of GBM patients with EGFRvIII (-) and (+) (p >0.05) (Figure 1A). The results suggest that the different expression of EGFRvIII may lead to the different biological characteristics of GBM cells, which was not related to important molecular typing markers of GBM.

10 Is the classification of the EGFRvIII group on the CGGA database dual, ergo only positive or negative, or there is information about the EGFRvIII expression?

Thank you for raising the question.

Transcriptome sequencing data and most molecular pathological information (MGMT status, Chromosome 1p/19q status, IDH status, and transcription subtypes) of glioma patients were obtained from CGGA official website (www.cgga.org.cn). The EGFRvIII status was obtained from the immunohistochemical staining of the above clinical samples.

11 I think the reference to selecting the PN and MES subtypes is not the most appropriate. Could you corroborate that CD133 is only expressed in the PN subtype? CD133 is a stem cell marker and might be expressed in the MES subtype too. Could the authors confirm that these genes are in the Verhaak or Wang signature?

Thank you for raising the question.

We also thought the markers selected as PN and MES subtypes may not the most appropriate, and CD133 is not only expressed in the PN subtype but also expressed the MES subtype. However, Mao et al. and Chandran et al. reported that SOX2, NOTCH1, OLIG2 and CD133 are the PN-associated genes, while CD44, BCL2A1, LYN and WT1 are the Mes-associated genes in PN and Mes GSCs [1,2]. Since there is no more suitable gene for the PN and MES subtypes markers, therefore, in the current study, we selected SOX2, NOTCH1, OLIG2 and CD133 as PN subtype marker and CD44, BCL2A1, LYN and WT1 as MES subtype marker.

  • Mao P, Joshi K, Li J, Kim SH, Li P, Santana-Santos L, Luthra S, Chandran UR, Benos PV, Smith L, Wang M, Hu B, Cheng SY, Sobol RW, Nakano I. Mesenchymal glioma stem cells are maintained by activated glycolytic metabolism involving aldehyde dehydrogenase 1A3. Proc Natl Acad Sci U S A. 2013 May 21;110(21):8644-9. doi: 10.1073/pnas.1221478110. Epub 2013 May 6. PMID: 23650391; PMCID: PMC3666732.
  • Chandran UR, Luthra S, Santana-Santos L, Mao P, Kim SH, Minata M, Li J, Benos PV, DeWang M, Hu B, Cheng SY, Nakano I, Sobol RW. Gene expression profiling distinguishes proneural glioma stem cells from mesenchymal glioma stem cells. Genom Data. 2015 Sep 1;5:333-336. doi: 10.1016/j.gdata.2015.07.007. PMID: 26251826; PMCID: PMC4523279.

12 Which number of replicates were used in each experiment? I suggest adding this information to confirm that the statistical analysis was correctly applied and the number of experiments is sufficient to obtain statistical significance. If there are fewer than three replicates and two cell lines used in any experiment, the authors should to performed more experiments to make the final conclusions.

Thank you for your suggestion.

We are sorry for not description of the number of replicates in each experiment. According to your suggestion, we have added the related description in the methods section (marked in yellow).

13 In addition, why the authors used only the two GBM cell lines and not the GSC for ALDH1A3 analysis? Has studied this topic in the third GBM cell line? In addition, has the authors studied the expression of ALDH1A3 in the CGGA database or human specimens? Are any relation in its expression with EGFRvIII expression?

Thank you for raising these questions.

We thought that the similar results may be obtained with the third GBM cell line, even so, we will also conduct the related experiments with a third cell line in the subsequent experiments to test our theory. Thank you for your suggestion. In addition, in the future, we will also investigate the expression of ALDH1A3 in the CGGA database or human specimens, and analyze the relationship between ALDH1A3 and EGFRvIII expression.

14 Which is the contribution of KEGG analysis to the study? Only in the KEGG analysis with the CGGA database, the pathways related to EGFRvIII are expressed. Is there another pathway related to PMT, for example, in the results of KEGG analysis with GBM cell lines? Are the authors comparing the KEEG analysis of the GBM cell line with EGFRvIII (+) Vs (-)? This could be more informative to compare the pathways activated with the overexpression of EGFRvIII.

Thank you for raising these questions.

In our study, KEGG analysis was performed on the second-generation transcriptome sequencing data to screen EGFRvIII related pathways. We screened the EGFRvIII related pathways in CGGA database and GBM cell lines respectively, and also analyzed their common pathway. The result of CGGA database showed that pathways related to EGFRvIII included cytokine-cytokine receptor interaction, chemical factor signaling pathway, NF-κB pathway, etc.. The results of GBM cell lines showed that compared with the control group, the EGFRvIII related pathways in GBM cell lines included cytokine-cytokine receptor interaction, tumor signaling pathway, etc.. NF-κB pathway is one of the important pathways in cancer. Further, we analyzed the different signaling pathways of the above two kinds of GBM cells by GSVA, and the results showed that 4 pathway enrichment scores decreased and 28 increased in U87-EGFRvIII (+) and LN229-EGFRvIII (+) cells, including the NF-κB pathway.

15 To confirm that the expression of EGFRvIII is involved in the NFkB more experiments and more cell lines must be used. How many experiments were used to obtain figure 5D-H?

Thank you for raising the question.

In the current study, we carried out the Western blot, ChIP experiment in U87-EGFRvIII (-) and (+) cells pretreatment with or without NF-κB inhibitor (JSH-23) to confirm the transcriptional regulation of NF-κB on ALDH1A3 gene in GBM cells. In addition, the bioinformatics analysis including LASAGNA software and motif analysis was used to predict the binding site of NF-κB and ALDH1A3 gene promoter region. Representative data in Figure 5D-H are determined from three independent experiments (n = 3 per group).

16 I think that in the legend of figure 6, there is some comment from some author or reviewer (sentences 485-387).

Thank you for raising the question.

We are sorry that some comments in the legend of Figure 6 were not deleted in time, and we have been deleted these comments from the text.

17 Finally, in the discussion section, E2F6 appears for the first time and there is not any related to this topic during the study. Is it important? Only there is this sentence, that's why I suggest deleting the sentence

Thank you for raising the question.

E2F6 was mentioned in the discussion section to illustrate that the NF-kB signaling pathway may also be involved in temozolomide resistance in GBM through other factors. We have deleted the sentence.

Once again, thank you very much for your comments and suggestions.

Sincerely yours,

Zhongfang Shi

*Corresponding authors: Xiaoguang Yu, Department of Biochemistry & Molecular Biology, Harbin Medical University, 157 Baojian Road, Harbin 150081, Heilongjiang, China. Email: [email protected].

Reviewer 3 Report

This manuscript is interesting and the text clearly explained.  Only a minor comment:

1.- GBM does not usually induce metastasis, this term should be either deleted (P1,L42) or explained. 

Author Response

Dear Reviewer:

Thank you for your comments concerning our manuscript entitled “EGFRvIII Promotes the Proneural-Mesenchymal Transition of Glioblastoma Multiforme and Reduces Its Sensitivity to Temozolomide by Regulating NF-κB/ALDH1A3 Axis”, (ID: genes-2203856).

These comments concerning our manuscript are all valuable and very helpful for revising and improving our paper, and have the important guiding significance for our future research. We have tried our best to revise our manuscript according to the comments. These revisions will not influence the content and framework of the paper. Revised parts are marked in yellow in the revised version. The main corrections in the paper and the responds to the reviewer's comments are as flowing:

Responds to the reviewer's comments:

This manuscript is interesting and the text clearly explained.  Only a minor comment:

1.- GBM does not usually induce metastasis, this term should be either deleted (P1,L42) or explained. 

Thank you for your suggestion.

We agree with you very much. Despite the common occurrence of GBM among primary brain tumors, metastasize outside of the CNS is rare because of the presence of the blood–brain barrier. GBM is characterized, in particular, by significant intratumor cell variability, high growth rates, and widespread invasiveness within the surrounding normal brain parenchyma. We have deleted this term “metastasis” from the text.

Once again, thank you very much for your comments and suggestions.

Sincerely yours,

Zhongfang Shi

*Corresponding authors: Xiaoguang Yu, Department of Biochemistry & Molecular Biology, Harbin Medical University, 157 Baojian Road, Harbin 150081, Heilongjiang, China. Email: [email protected].

Round 2

Reviewer 2 Report

The suggestions have been answered satisfactorily. More experiments with more cell lines could be done but these might be done in future studies since the bioinformatic analysis corroborate the experimental results and all experiments are correctly correlated.